# Chiral aldehyde-nickel dual catalysis enables asymmetric α−propargylation of amino acids and stereodivergent synthesis of NP25302

Fang Zhu[1], Chao-Xing Li[1], Zhu-Lian Wu [1], Tian Cai [1], Wei Wen [1] ✉ & Qi-Xiang Guo [1] ✉

The combined catalytic systems derived from organocatalysts and transition metals exhibit powerful activation and stereoselective-control abilities in asymmetric catalysis. This work describes a highly efficient chiral aldehyde-nickel dual catalytic system and its application for the direct asymmetric α−propargylation reaction of amino acid esters with propargylic alcohol derivatives. Various structural diversity α,α−disubstituted non-proteinogenic α−amino acid esters are produced in good-to-excellent yields and enantios-electivities. Furthermore, a stereodivergent synthesis of natural product NP25302 is achieved, and a reasonable reaction mechanism is proposed to illustrate the observed stereoselectivity based on the results of control experiments, nonlinear effect investigation, and HRMS detection.

The development of catalytic systems is an important work for asymmetric catalysis[1,2]. One of the most active research fields is the development of combining catalytic systems from organocatalysts and transition metals[3–8]. For example, chiral organocatalysts such as quaternary ammonium salts[9,10], amines[11–17], and brønsted acids[18–22] have already been proven excellent catalyst partners for transition metals including palladium, rhodium, iridium, ruthenium, nickel, copper, etc. With the utilization of these combined catalysts, numerous challenging asymmetric organic transformations were achieved[3–8]. As an emerging asymmetric catalytic strategy, chiral aldehyde catalysis has been proven the most preferred one for the direct asymmetric α−functionalization of *N*-unprotected amino-methyl compounds[23–27]. However, most of the reported examples focused on the usage of chiral aldehydes as pure organocatalysts[28–35], and the chiral aldehyde/transition metal combining catalytic systems were very rare (Fig. 1a)[36–38]. Especially, there was only one type of transition metal, the palladium, has been merged with chiral aldehyde catalysts[36–38]. So, the development of chiral aldehyde/transition metal-involved combining catalytic systems becomes an important

way to achieve more challenging α−functionalization reactions of aminomethyl compounds.

As a continuous work on our discovery of the chiral aldehyde/palladium combined catalysis[36–38], we tried to employ another type of transition metal as a catalyst partner with chiral aldehydes. Palladium is a soft metal that has a large atomic radius and strong electronegative property, while nickel has different chemical properties (harder, smaller atomic radius, and less electronegative)[39–42]. Due to these unique properties, nickel catalysis has been widely used in organic reactions such as cross-coupling[43,44], C-H activation[45,46], reductive coupling[47,48], etc. Among those reactions, the nickel-catalyzed asymmetric propargylation is an important strategy for the construction of optically active alkyne compounds[49–58]. Especially, the chiral Ni/Cu dual catalyzed asymmetric α−propargylation of aldimine esters reported by Guo et al provided a good solution for the preparation of chiral propargyl-functionalized amino acids[59,60], a type of useful compound that has been seldom studied by synthetic chemists (Fig. 1b)[61–65]. With consideration of the unique properties of chiral aldehyde catalysis in activating amino acid derivatives, the combined

[1]Key Laboratory of Applied Chemistry of Chongqing Municipality, and Chongqing Key Laboratory of Soft-Matter Material Chemistry and Function Manufacturing, School of Chemistry and Chemical Engineering, Southwest University, Chongqing 400715, China. ✉e-mail: wenwei1989@swu.edu.cn; qxguo@swu.edu.cn

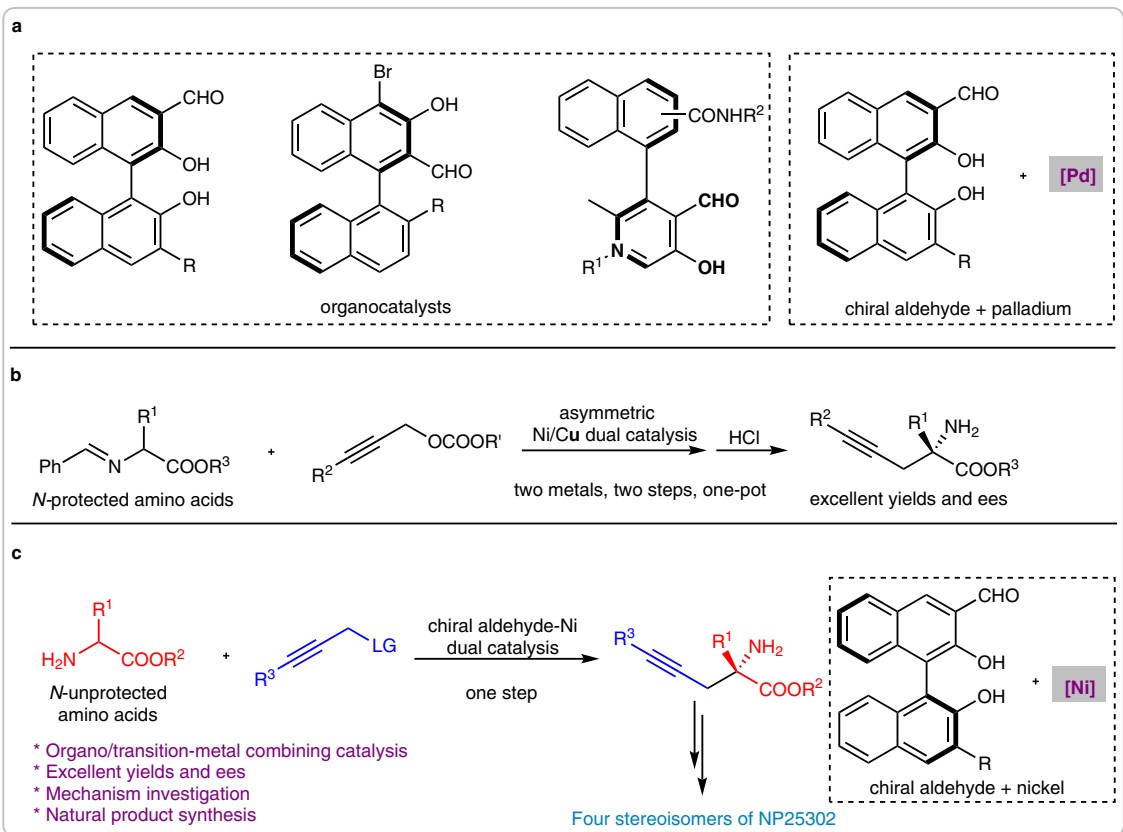

**Fig. 1 | The reported chiral aldehyde catalytic systems and the asymmetric α−propargylation reaction of amino acid esters (LG = Leaving Group). a** The reported chiral aldehyde-involved catalytic systems. **b** The reported catalytic asymmetric α−propargylation of amino acid esters. **c** The chiral aldehyde/nickel catalyzed α−propargylation of N-unprotected amino acid derivatives (this work).

chiral aldehyde/nickel catalytic system is fully anticipated for achieving this important transformation without additional steps of protection and deprotection.

In this work, we rational design a chiral aldehyde/nickel combining catalytic system, which can efficiently promote the direct asymmetric α−propargylation reaction of N-unprotected amino acid esters with propargyl alcohol derivatives. Various propargyl-functionalized α,α−disubstituted α−amino acid esters are generated in good-to-excellent yields and enantioselectivities. With the utilization of control experiments, nonlinear effect investigation and HRMS detection, a reasonable mechanism is proposed. Furthermore, this method is used for the stereodivergent synthesis of natural product NP25302 (Fig. 1c).

## Results

### Optimization of reaction conditions

Our work initiated with the evaluation of the reaction between *tert*-butyl alaninate **1a** and benzoyl-protected 3-phenyl propargyl alcohol ester **2a**, in the promotion of a combined catalytic system of chiral aldehyde **3a** and Ni(COD)$_2$. Lewis acid ZnCl$_2$ and base 1,1,3,3-tetramethylguanidine (TMG) were added to accelerate the successive processes of Schiff base formation and deprotonation. As expected, the desired product **4a** was generated with moderate yield and excellent enantioselectivity (Table 1, entry 1). Then, a series of reaction condition optimizations were carried out. Leaving group screening indicated that OAc was the optimal one that could give product **4a** in 49% yield and with 94% ee (Table 1, entry 2). Base screening showed that TMG was the best choice (Supplementary Table 1). After we tuned the equivalents of the base from 1.0 to 1.6, the yield of **4a** was efficiently improved to 71% (Table 1, entry 6; Supplementary Table 2). Subsequently, various diphosphine ligands were examined (Supplementary Table 3). Among them, chiral ligand **L4** gave the best yield, albeit the

enantioselectivity of **4a** decreased slightly (Table 1, entry 9). With the utilization of TMG as base and **L4** as ligand, chiral aldehydes **3b-3k** and achiral aldehyde **3l-3m** were individually employed as cocatalysts to replace chiral aldehyde **3a**. We found that the combination of chiral aldehyde **3i** and ligand **L4** gave the best experimental results (Table 1, entry 17). The reactant concentration affected the yield of **4a** slightly. After we increased the concentration of reactant **1a** from 0.2 M to 0.4 M, the yield of **4a** was further improved to 92% (Table 1, entry 22). Finally, the matching relationship between the chiral aldehyde and chiral ligand was investigated. Results show that the combination of *ent*-**3i** and **L4** gave product **4a** merely in 25% yield and 62% ee (Table 1, entry 23). So, the reaction conditions in entry 22 were chosen as the optimal ones for the next substrate scopes investigation. Under these optimal reaction conditions, we replaced the metal Ni(COD)$_2$ with palladium Pd(PPh$_3$)$_4$ or [Pd(C$_3$H$_5$)Cl]$_2$, no desired reaction occurred (Table 1, entry 24). These results indicated that this chiral aldehyde/nickel combining catalytic system has unique properties in achieving this type of organic transformation.

### Substrate scope of propargylic alcohol derivatives

With the optimal reaction conditions in hand, we then investigated the substrate scopes. Firstly, various propargylic alcohol derivatives were tested. 3-Phenylprop-2-yn-1-yl acetates were good reaction partners for amino acid ester **1a**, and the yields varied with the change of substituent position. Moderate yields were observed when compounds **2** bearing an *ortho*-substituted phenyl were involved in this reaction (Fig. 2, **4b-d**). These moderate yields were possibly caused by the steric effect of *ortho*-substituents. Once the substituent was installed at the *meta*- or *para*-position of the phenyl, the desired products were obtained in good-to-excellent yields (Fig. 2, **4e-4m**). Similar yields were observed in the reactions of **1a** with corresponding

**Table 1 | Reaction condition optimization**

| Entry | 3 | Ligand | LG | Yield (%)[a] | ee (%)[b] |
|---|---|---|---|---|---|
| 1 | **3a** | **L1** | OBz | 46 | 95 |
| 2 | **3a** | **L1** | OAc | 49 | 94 |
| 3 | **3a** | **L1** | OBoc | 34 | 90 |
| 4 | **3a** | **L1** | OCO$_2$Me | 18 | 74 |
| 5 | **3a** | **L1** | OPO(OEt)$_2$ | 14 | 76 |
| 6[c] | **3a** | **L1** | OAc | 71 | 93 |
| 7[c] | **3a** | **L2** | OAc | 6 | 22 |
| 8[c] | **3a** | **L3** | OAc | 10 | 87 |
| 9[c] | **3a** | **L4** | OAc | 80 | 91 |
| 10[c] | **3b** | **L4** | OAc | 64 | 97 |
| 11[c] | **3c** | **L4** | OAc | 11 | 94 |
| 12[c] | **3d** | **L4** | OAc | 70 | 94 |
| 13[c] | **3e** | **L4** | OAc | 78 | 90 |
| 14[c] | **3 f** | **L4** | OAc | 63 | 93 |
| 15[c] | **3 g** | **L4** | OAc | 14 | 94 |
| 16[c] | **3 h** | **L4** | OAc | 83 | 96 |
| 17[c] | **3i** | **L4** | OAc | 87 | 96 |
| 18[c] | **3j** | **L4** | OAc | 9 | 40 |
| 19[c] | **3k** | **L4** | OAc | 0 | – |
| 20[c] | **3 l** | **L4** | OAc | 33 | 82 |
| 21[c] | **3 m** | **L4** | OAc | 0 | – |
| 22[cd] | **3i** | **L4** | OAc | 92 | 96 |
| 23[cd] | ent-**3i**[e] | **L4** | OAc | 25 | 62 |
| 24[cdf] | **3a** | **L1** | OAc | trace | – |

[a]Isolated yield.
[b]Determined by chiral HPLC.
[c]With 160 mol% TMG.
[d]With 0.5 mL PhCH$_3$.
[e]ent-**3i** = the enantiomer of chiral aldehyde **3i**.
[f]Ni(COD)$_2$ was replaced by Pd(PPh$_3$)$_4$ or [Pd(C$_3$H$_5$)Cl]$_2$.

propargylic alcohol derivatives bearing 3,4-disubstituted phenyl units (Fig. 2, **4n-4p**). Notably, all of these 3-phenylprop-2-yn-1-yl acetates gave products with excellent enantioselectivities (92–98% ee). Other aryl-substituted propargylic alcohol acetates, including 2-naphthyl, 1-naphthyl, 9H-fluoren-3-yl, and 3-thienyl, were then tested. Except for that the 1-naphthyl substituted propargylic alcohol acetate gave product **4r** in moderate yield (58%), all others reacted efficiently with **1a** and gave desired products in excellent yields and enantioselectivities (Fig. 2, **4q-4t**). Saturated aliphatic alkyl-substituted propargylic alcohol acetates also exhibited high reactivity with amino acid ester **1a**, giving products **4s-4v** in excellent yields and enantioselectivities. Two 3-phenylprop-2-yn-1-yl acetates bearing chiral side chains were tested; products **4w** and **4x** were obtained with moderate yields and excellent stereoselectivities (>20:1 dr). We found the secondary propargylic

alcohol ester could not efficiently react with **1a** under the optimal reaction conditions (Fig. 2, **4aa**).

**Substrate scope of amino acid esters**

Next, the substrate scope of amino acid esters was investigated. Phenyl glycine esters could participate in this reaction efficiently, however, to obtain high yields, it was necessary to increase the chiral aldehyde catalyst loading and rise the reaction temperature (Fig. 3, **5a-5d**). Phenylalanine and homophenylalanine-derived esters also reacted efficiently with **2b**, leading to products **5e-5i** in good yields and excellent enantioselectivities. Representative amino acid esters bearing aliphatic alkyl, allyl, sulfur and ester-containing alkyls were used as donors, and all of them gave desired products in good-to-excellent yields and enantioselectivities (Fig. 3, **5j-5n**).

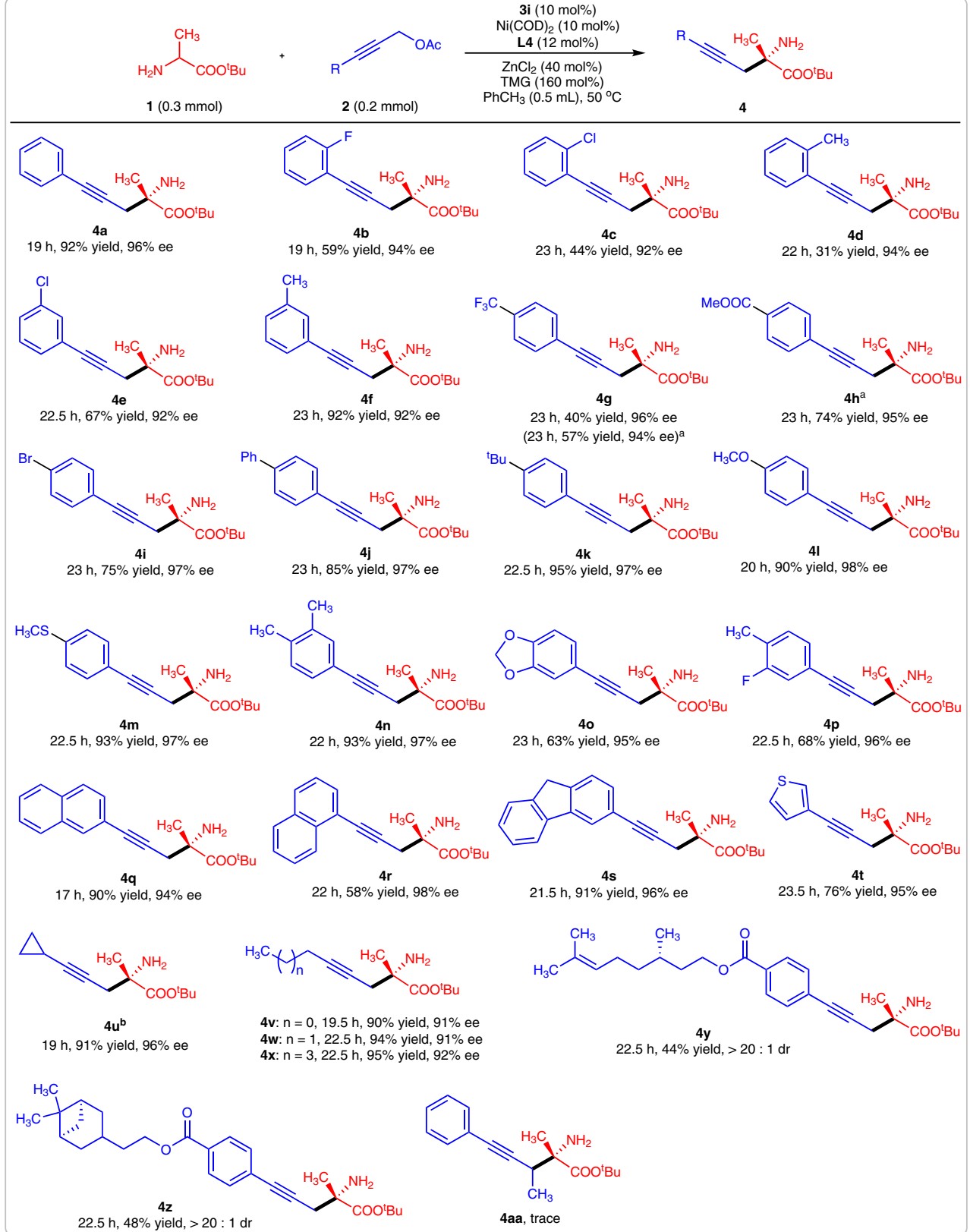

**Fig. 2 | The substrate scope of propargylic alcohol derivatives. a** With 20 mol% **3i** and at 60 °C. **b** Ee value was obtained from its N-Bz-protected derivative.

## Stereodivergent synthesis of NP25302

NP25302 is a natural pyrrolizidine alkaloid that shows excellent biological activity in inhibiting the adhesion of HL-60 cells to CHO-ICAM-1 cells (IC$_{50}$ = 27.2 µg/mL)[66]. However, studies on the total synthesis of

this compound were very limited. In 2006, Snider and co-workers described a total synthesis of (*R, R*)- and (*S, S*)-NP25302[67]. Subsequently, Robertson and co-workers achieved a total synthesis of this compound in a racemic manner[68]. There are two chiral centers in

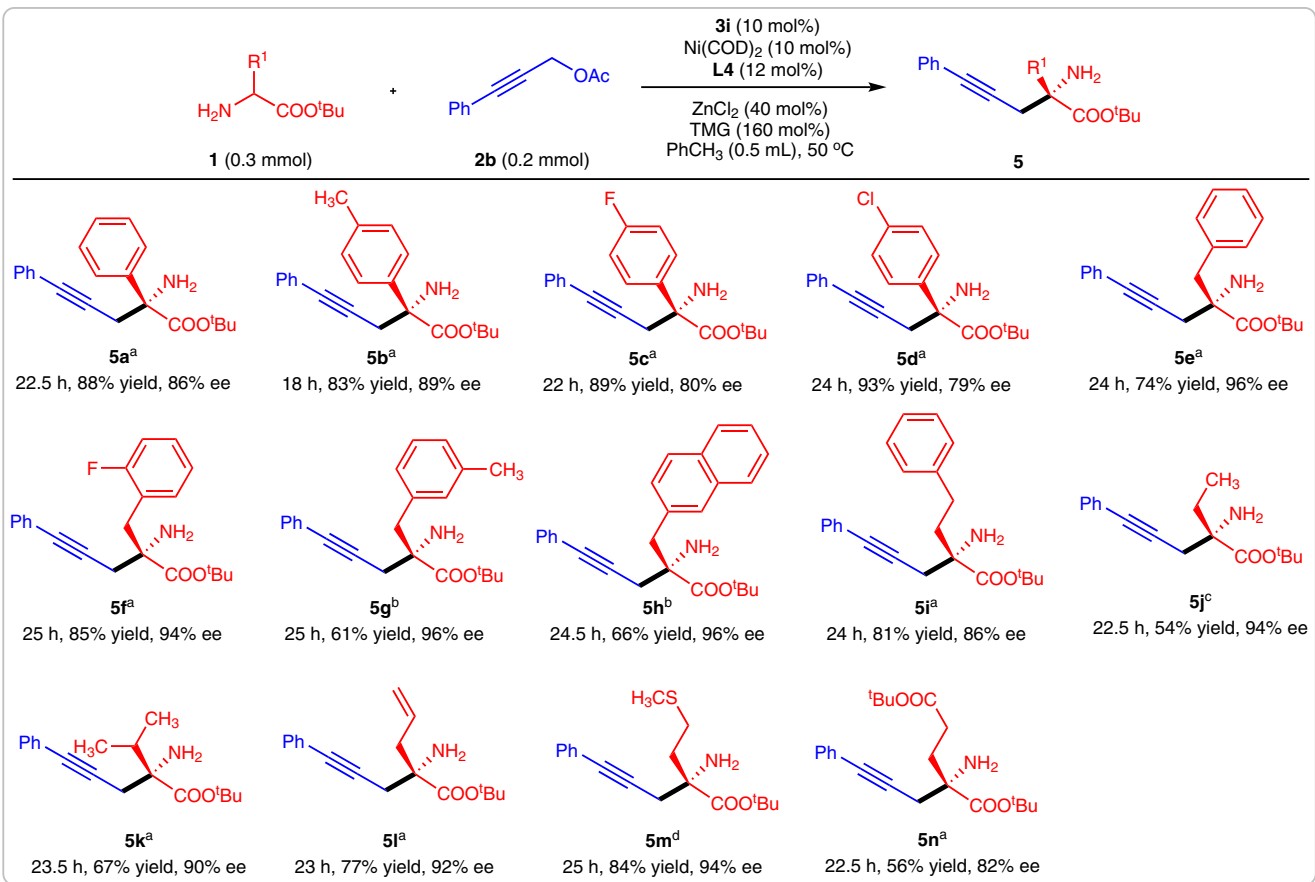

**Fig. 3 | The substrate scope of amino acid esters. a** With 20 mol% **3i** and at 60 °C. **b** With 20 mol% **3a** and at 60 °C. **c** With 20 mol% **3i**, 12 mol% **L1**, and at 60 °C. **d** With 12 mol% **L1**.

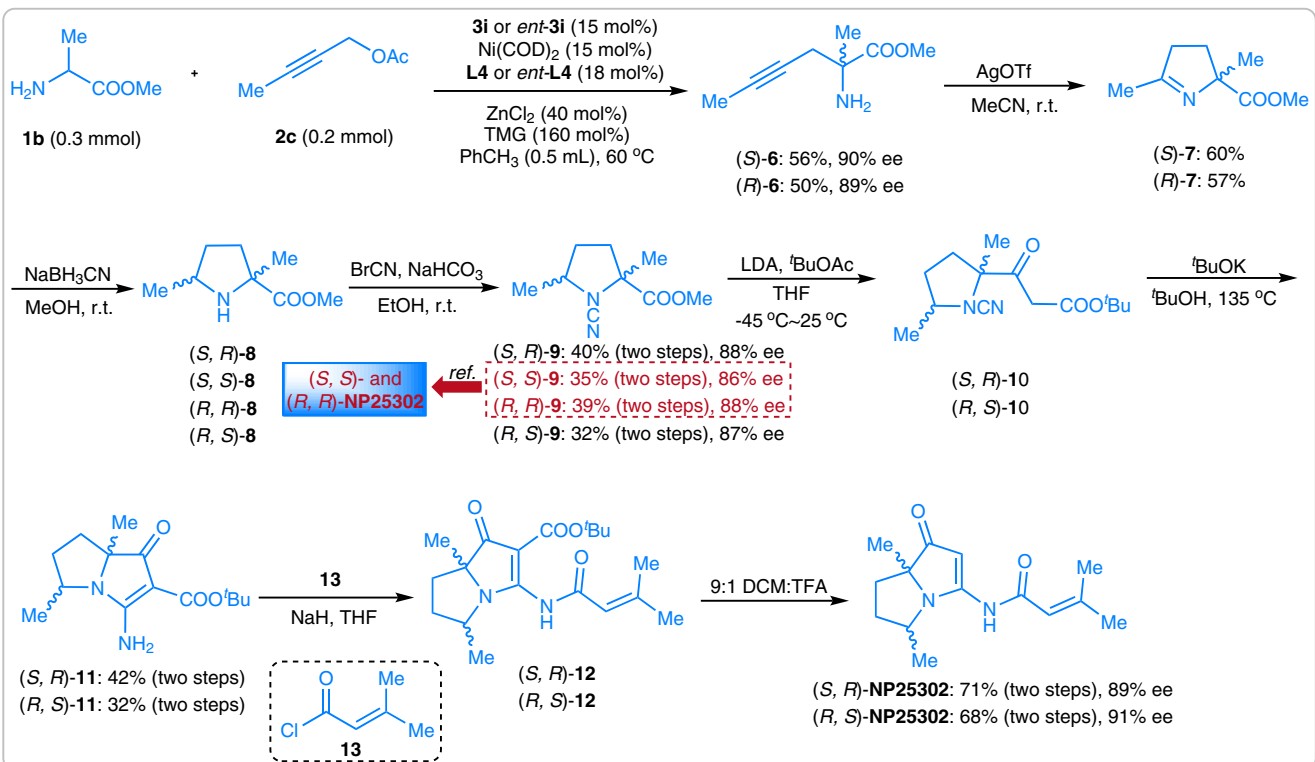

**Fig. 4 | Synthetic application.** The stereodivergent synthesis of **NP25302** from amino acid ester **1b** and propargylic alcohol ester **2c**.

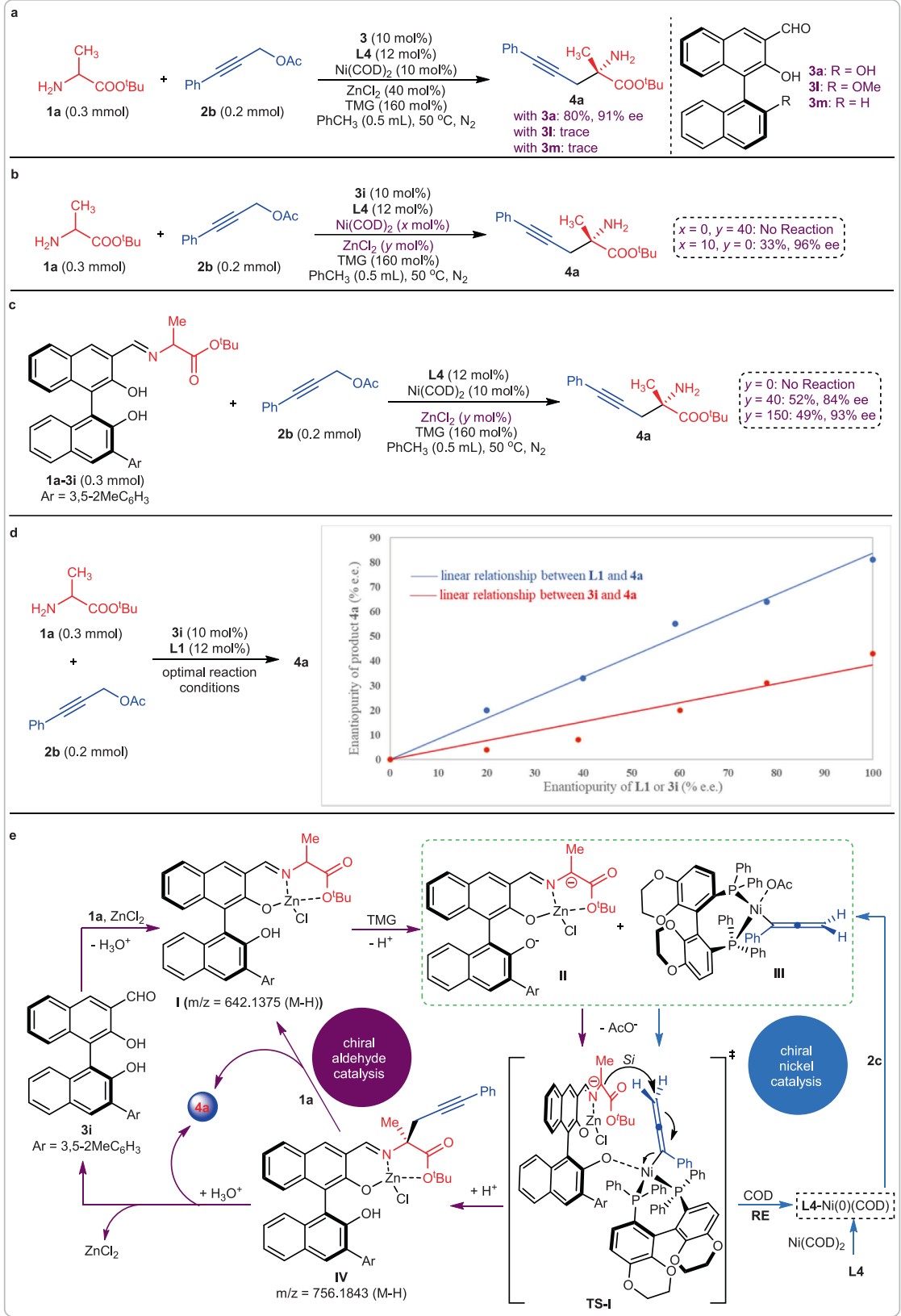

**Fig. 5 | Reaction mechanism investigation. a** Control experiments with modified chiral aldehyde catalysts. **b** Control experiments to investigate the role of nickel and ZnCl₂. **c** Control experiments with Schiff base as reactants. **d** The nonlinear effect investigation between the ee value of product **4a** and chiral sources **3i** or **L1**. **e** The possible catalytic cycles. (x = the equivalents of Ni(COD)₂, y = the equivalents of ZnCl₂).

this molecule, but the attempt to achieve all of the four stereoisomers has never been touched. With chiral aldehyde **3i** and ligand **L4**, the methyl alaninate **1b** reacted with but-2-yn-1-yl acetate **2c** smoothly giving (*S*)-**6** in 56% yield and 90% ee. Corresponding (*R*)-**6** was obtained under the promotion of chiral aldehyde *ent*-**3i** and ligand *ent*-**L4** (enantiomer of **L4**). Treatment of compounds **6** with AgOTf produced dihydropyrroles **7** in moderate yields. Then, the imine groups of **7** were reduced by NaBH$_3$CN. The (*S, R*)-**8** and (*S, S*)-**8** were generated from (*S*)-**7**, and the (*R, R*)-**8** and (*R, S*)-**8** were generated from (*R*)-**7**. All of the diastereoisomers were separated by flash column chromatography. After reacting with BrCN under basic conditions, four stereoisomers of **9** were obtained in good yields. (*S, S*)- and (*R, R*)-**9** were reported as the key chiral building blocks leading to (*S, S*)- and (*R, R*)-NP25302[67]. So, (*S, R*)- and (*R, S*)-**9** were used for the synthesis of the other two isomers. (*S, R*)- and (*R, S*)-**11** were produced in good yields by a Claisen condensation with *tert*-butyl acetate and an intramolecular cyclization reaction. Protecting the amino group of compounds **11**, and decarboxylation of their *tert*-butyl ester groups, (*S, R*)- and (*R, S*)-NP25302 were respectively generated in good yields and enantioselectivities. Thus, all four stereoisomers of NP25302 could be synthesized from the readily available starting materials **1b** and **2c** within 8 steps (Fig. 4).

## Discussion

The possible reaction mechanism was then investigated. Firstly, two modified chiral aldehyde catalysts **3n** and **3o** were used to promote the model reaction, and only trace amounts of **4a** were observed. Comparing these results with that obtained from chiral aldehyde catalyst **3a** shows that the 2′-hydroxyl is vital for this reaction (Fig. 5a). Like the transition state we previously disclosed[8], the formation of a coordination bond between the 2′-hydroxyl and an active nickel species is possible. The role of the transition metal and Lewis acid were then studied. In the absence of nickel, this reaction could not take place, indicating that the active electrophile intermediate must be generated by nickel catalysis. The yield of **4a** decreased greatly in the absence of ZnCl$_2$, showing that the reaction process could be accelerated by this Lewis acid. The reaction of Schiff base **1a**-**3i** with **2b** was carried out to further investigate the role of Lewis acid. Results indicated that the yield and ee varied with the equivalents of ZnCl$_2$. No reaction occurred in the absence of ZnCl$_2$. When 0.4 equivalents of Lewis acid were employed, product **4a** was generated in 52% yield and 84% ee. After the equivalents of Lewis acid were increased to 1.5, the ee of **4a** was enhanced to 93% (Fig. 5c). These results indicated that the Lewis acid ZnCl$_2$ was involved in the transition state and acted a vital role in the enantioselective control. To verify this, we detected this reaction by HRMS. Two Schiff base-Zn complexes, **I** and **IV**, were directly observed and furtherly verified by comparing their isotopic distributions with theoretical data (Supplementary Fig. 3). All of these results provided good evidence for the conclusion that the Lewis acid ZnCl$_2$ could: (1) speed the Schiff base formation process, (2) furtherly enhance the α−carbon acidity of the Schiff base and then accelerate the subsequent deprotonation process, and (3) strengthen the stereoselective-control ability of the transition state. The nonlinear effect of the enantiopurity between product **4a** and two chiral sources, the chiral aldehyde **3i** and chiral ligand **L1**, was then studied. Results indicated that both these two chiral sources exhibited linear relationships with **4a**, so, it is reasonable to deduce that only one molecule of chiral aldehyde and one molecule of chiral ligand were involved in the transition state (Fig. 5d). Combining the above results with the absolute *S* conformation of product **4a**, a possible reaction mechanism was proposed in Fig. 5e. Chiral aldehyde **3i** reacted with amino acid ester **1a** in the promotion of Lewis acid ZnCl$_2$, leading to the stable Schiff base-Zn complex **I**. Then this complex was deprotonated by TMG to form an active nucleophile **II**. At the same time, an active chiral nickel species (**III**) was formed from propargylic alcohol ester **2c** and **L4**-Ni(0)(COD) via oxidative addition. With a ligand

exchange, **TS-I** was formed from active intermediates **II** and **III**. The α−carbon anion of **II** provided its *Si* face to attack the active allenylic nickel species, leading to Schiff base-Zn complex **IV** via reductive elimination (**RE**) and protonation processes. Product **4a** was then generated by hydrolysis or amine exchange.

In conclusion, we disclosed a highly efficient chiral aldehyde-nickel dual catalytic system and its application in the asymmetric α−propargylation reaction of *N*-unprotected amino acid esters with propargylic alcohol derivatives. Forty-two structural diversity α,α−disubstituted α−amino acid esters were obtained in yields of 31−95% and ee values of 79−98%. Products (*R*)- and (*S*)-**6** were used for the total synthesis of the four stereoisomers of natural product NP25302. According to the results given by control experiments and nonlinear effect investigation, and the key intermediates detected by HRMS, a reasonable reaction mechanism is proposed to illustrate the enantioselective control phenomenon.

## Methods

### Method for the catalytic asymmetric α−propargylation of amino acids

In a nitrogen-filled glove box, an oven-dried 10 mL screw-cap reaction tube equipped with a stir bar was charged with Ni(COD)$_2$ (5.5 mg, 0.02 mmol), (*R*)-synphos (15.3 mg, 0.024 mmol) and stirred in toluene (0.5 mL) at room temperature for about 5 minutes. Then, *tert*-butyl amino acid ester **1** (0.3 mmol), propargylic acetate ester **2** (0.2 mmol), chiral aldehyde **3i** (8.2 mg, 0.02 mmol), ZnCl$_2$ (10.9 mg, 0.08 mmol) and TMG (36.8 mg, 0.32 mmol) were added. The mixture was continuously stirred at 50 °C under a nitrogen atmosphere. After the reaction was completed, the solvent was removed by rotary evaporation, and the residue was purified by flash chromatography separation on a silica gel column (eluent: petroleum ether/ethyl acetate/triethylamine = 250/100/2). The details of the full experiments and compound characterizations were provided in the Supplementary Information.

## Data availability

The authors declare that all other data supporting the findings of this study are available within the article and its Supplementary Information file.

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

## Acknowledgements
We are grateful for financial support from NSFC (22071199, 22201235), the Innovation Research 2035 Pilot Plan of Southwest University (SWU-XDZD22011), and the Chongqing Science Technology Commission (cstccxljrc201701, cstc2018jcyjAX0548).

## Author contributions
W.W. and G.Q.X. conceived this project. Z.F. and L.C.X. carried out the experiments. W.Z.L. and C.T. performed the HRMS analysis. G.Q.X. wrote the manuscript. All authors discussed the results.

## Competing interests
The authors declare no competing interests.
