## [Peer Review File · Nature Communications]

REVIEWER COMMENTS

Reviewer #1 (Remarks to the Author):

In this manuscript, Guo and co-workers reported a chiral aldehyde-nickel dual catalytic system and its application in the asymmetric propargylation reaction of amino acid esters and propargylic alcohol derivatives. This methodology could be applied in the stereodivergent synthesis of natural product NP25302. Although a similar strategy has been employed in the cooperative Ni/Cu-catalyzed propargylic alkylation by the group of Guo (Angew. Chem. Int. Ed. 2020, 59, 14270 & Nat. Synth. 2022, 1, 393), the present work can achieve this kind of transformation by using single metal catalyst-Ni and without additional amino protection/deprotection steps. In this context, I recommend its publication on Nat. Commun. after minor revision as follows:

1. the author should check carefully for spelling and grammar mistakes. For example, the title “for The Asymmetric α -Propargylation” should be changed to “for the Asymmetric α -Propargylation”.

The “optimization of reaction conditins” should be changed to “optimization of reaction conditions”.

“Base screening showed that TMG was the base of choice (Supporting Information, Table S1)” should be changed to “Base screening showed that TMG was the best of choice (Supporting Information, Table S1)”

2. The description “only one metal is involved in the catalytic system” is inappropriate as ZnCl₂ was used.

3. The substrate scope of propargylic alcohol derivatives was insufficient to support the protocol's functional group tolerance. There is no example for the substrates bearing electron-withdrawing groups, such as trifluoromethyl, ester, or cyano groups on the aromatic ring. Besides, the substrate scope of propargylic alcohol derivatives was limited to primary propargylic alcohol derivatives. Can secondary and tertiary propargylic alcohol derivatives be employed as the substrates?

4. In the mechanism studies, the exact role of Lewis acid ZnCl₂ is unclear. There are several possibilities: 1) Lewis acid ZnCl₂ can accelerate the condensation of amine and aldehyde, 2) ZnCl₂ can promote deprotonation by TMG to form an active nucleophile. 3) ZnCl₂ can promote the leaving of OAc in compound 2c in Figure 5d. I suggest the authors design proper control experiments to verify its exact roles.

Reviewer #2 (Remarks to the Author):

In this manuscript, Wen and Guo presented the chiral aldehyde/nickel dual catalytic system for the asymmetric α -propargylation of N-protected amino acid esters with propargyl alcohol derivatives. Various α,α -disubstituted α -amino acid esters were synthesized with good to high yields and high enantioselectivities. Both aromatic and aliphatic substituted propargylic alcohol acetates were demonstrated. The authors have performed mechanistic experiments, including control experiments and a nonlinear effect study, which reveals the vital role of the 2'-hydroxyl group of chiral aldehyde in this reaction. They also detect the key intermediates by HRMS analysis. The authors have proposed a possible reaction mechanism that shows a crucial role of ZnCl₂ in this catalytic process. The authors have also synthesized four stereoisomers of natural product NP25302. A similar methodology to synthesize chiral propargyl-functionalized amino acids has been demonstrated by Guo using Ni/Cu bimetallic system, though protected amino acids were employed. In fact, in the current method, the employed aldehydes are used as a transient protecting group, which requires an additional ZnCl₂ precursor. Moreover, the same group extensively demonstrated the original concept of "chiral aldehyde/metal combined catalysis" for the synthesis of diverse molecules (ref 36-38), and other groups have demonstrated the use of propargylic electrophiles in the reactions (ref 50-59). Considering all the precedents, the present work conceptually lacks the novelty to be considered for publication in the journal Nat. Commun.

Numerous errors in the manuscript should be addressed:

- 1) In the abstract and elsewhere, modify the sentences as ".....the asymmetric propargylation reaction of amino acid esters with (not and) propargylic alcohol....."
- 2) Reaction optimization: why does the product yield decrease upon increasing the concentration of TMG (Table S2, entry 5)? Similarly, in solvent screening, the reaction in toluene gave the 71% yield, while o-xylene and mesitylene gave only 4% yield (Table S7, entries 4-6)
- 3) What is "ent-3i"? Clearly define the same. Reaction time is not mentioned in Table 1 or the footnotes.
- 4) Table 1, entry 22: reaction should have been checked with a Pd(0) precursor, as Ni(0) is used in the original reaction. Moreover, the formula of Pd(II) species in the footnotes of Table 1 is incorrect.
- 5) Figure 4: Please mention the reaction partners from 9 to 10 and 10 to 11
- 6) Page 2: ...unique priority of chiral aldehyde....the word "priority" seems odd here.
- 7) The authors should provide ¹⁹F NMR spectra for fluorinated compounds.

8) In Table 1, the molecular formula for chiral aldehyde 3g is incorrect.

9) On page no 6, line no 1, the spelling of propargylic is incorrect.

10) References 45 and 47 are the same.

Reviewer #3 (Remarks to the Author):

Guo, Wen and co-workers described a chiral aldehyde/nickel catalytic system and its application in the asymmetric α -propargylation reactions of amino acid esters and propargyl alcohol derivatives. The development of novel combining catalytic systems from organocatalysts and transition metals is one of the most active research fields in asymmetric catalysis. Although chiral aldehyde catalysts exhibited powerful activation and excellent stereoselective-control abilities in the amino acids-involved reactions, examples of combining catalysis derived from chiral aldehydes and transition metals are very limited. Based on their research on the chiral aldehyde/palladium catalysis, the authors employed the transition metal nickel as catalyst partner for chiral aldehyde in this work. Due to the different chemical properties between palladium and nickel, chiral aldehyde/nickel catalysis is anticipated to realize a wide range of new reactions that cannot be achieved by previously reported catalytic systems, such as the asymmetric propargylation reaction disclosed in this work. Furthermore, the methodology disclosed here has broad substrate scopes and excellent stereoselective outcomes, and the application of this methodology was well documented by the stereodivergent synthesis of natural product NP25302. So, this reviewer recommended this work to be accepted by Nature Communications. Before acceptance, the following issues should be addressed.

1) Two chiral sources, the chiral aldehyde and chiral ligand, were used in this reaction. Ligand screening indicated that chiral diphosphine is necessary for this reaction, but no results to illustrate the necessity of chiral aldehyde. A combining catalytic system derived from achiral aldehyde and chiral palladium should be examined in this reaction.

2) In the supplementary information, the HPLC spectrums of compounds 6 and 9 showed that there existed a lot of impurities. So these two compounds should be further purified and corresponding data of yields should be modified.

Response Letter

Dear referees,

Thank you very much for your valuable and expert comments on our work. Here, we carefully revised our manuscript according to your comments and provided our responses and details of revisions in this letter.

1. Response and revisions to Reviewer #1.

Main comments: In this manuscript, Guo and co-workers reported a chiral aldehyde-nickel dual catalytic system and its application in the asymmetric propargylation reaction of amino acid esters and propargylic alcohol derivatives. This methodology could be applied in the stereodivergent synthesis of natural product NP25302. Although a similar strategy has been employed in the cooperative Ni/Cu-catalyzed propargylic alkylation by the group of Guo (Angew. Chem. Int. Ed. 2020, 59, 14270 & Nat. Synth. 2022, 1, 393), the present work can achieve this kind of transformation by using single metal catalyst-Ni and without additional amino protection/deprotection steps. In this context, I recommend its publication on Nat. Commun. after minor revision as follows:

Question 1. the author should check carefully for spelling and grammar mistakes. For example, the title “for The Asymmetric α -Propargylation” should be changed to “for the Asymmetric α -Propargylation”. The “optimization of reaction conditins” should be changed to “optimization of reaction conditions”. “Base screening showed that TMG was the base of choice (Supporting Information, Table S1)” should be changed to “Base screening showed that TMG was the best of choice (Supporting Information, Table S1)”

Our response and revisions: Thanks for your suggestion very much. Sorry for our carelessness. We thoroughly checked the whole manuscript. All of the issues mentioned in this question and other mistakes in spelling and grammar were corrected carefully.

Question 2. The description “only one metal is involved in the catalytic system” is inappropriate as $ZnCl_2$ was used.

Our response and revisions: Thanks for your comments very much. This description was not appropriate here. We changed this description to ‘organo/transition-metal combining catalysis’ in our Revised Manuscript.

Question 3. The substrate scope of propargylic alcohol derivatives was insufficient to support the protocol's functional group tolerance. There is no example for the substrates bearing electron-withdrawing groups, such as trifluoromethyl, ester, or cyano groups on the aromatic ring. Besides, the substrate scope of propargylic alcohol derivatives was limited to primary propargylic alcohol derivatives. Can secondary and tertiary propargylic alcohol derivatives be employed as the substrates?

Our response and revisions: Thanks for your suggestion very much. We added two substrates bearing electron-withdrawing groups ($-CF_3$ and $-COOMe$) in our Revised Manuscript (see Figure 2, **4g** and **4h**). For the secondary propargylic alcohol ester, no desired reaction occurred under our optimal reaction conditions, we also added this result in our Revised Manuscript (see Figure 2, **4aa**).

Question 4. In the mechanism studies, the exact role of Lewis acid $ZnCl_2$ is unclear. There are several possibilities: 1) Lewis acid $ZnCl_2$ can accelerate the condensation of amine and aldehyde, 2) $ZnCl_2$ can promote deprotonation by TMG to form an active nucleophile. 3) $ZnCl_2$ can promote the leaving of OAc in compound **2c** in Figure 5d. I suggest the authors design proper control experiments to verify its exact roles.

Our response and revisions: Thanks for your suggestion very much. We carried out several control experiments in this revision.

1) The reaction of amino acid ester **1a** and propargylic alcohol ester **2b** could proceed in the absence of $ZnCl_2$, but the yield decreased greatly (Figure 5b in Revised Manuscript). This result indicated that the Lewis acid $ZnCl_2$ can accelerate this reaction. So, the first opinion that Lewis acid $ZnCl_2$ can accelerate the condensation of amine and aldehyde is reasonable.

2) The reaction of amino acid esters **1a** and propargylic alcohol ester **2b** could not proceed in the absence of nickel (Figure 5b in Revised Manuscript). This result indicated that the Lewis acid $ZnCl_2$ could not promote the leaving of OAc in this reaction, and nickel is the true catalyst for the generation of active electrophile intermediate.

3) The reaction of Schiff base **1a-3i** (prepared from chiral aldehyde **3i** and amino acid ester **1a**) with propargylic alcohol ester **2b** could not proceed in the absence of $ZnCl_2$. However, after we increase the equivalents of $ZnCl_2$, this reaction could give the product in moderate yields, and ee values of the product varied with the equivalent of $ZnCl_2$ (Figure 5c in Revised

Manuscript). These results may be indicated that the Lewis acid ZnCl_2 could enhance the α -carbon acidity of the Schiff base via the formation of the Zn-Schiff base complex, and then accelerate the deprotonation process. Furthermore, this Lewis acid was involved in the transition state and could affect the enantioselectivity in some degree. The variation in yields may be caused by the equivalent ratio of Lewis acid and Schiff base.

All of the above control experiments and corresponding descriptions were added to our Revised Manuscript.

2. Response and revisions to Reviewer #2

Main comments: In this manuscript, Wen and Guo presented the chiral aldehyde/nickel dual catalytic system for the asymmetric α -propargylation of N-protected amino acid esters with propargyl alcohol derivatives. Various α,α -disubstituted α -amino acid esters were synthesized with good to high yields and high enantioselectivities. Both aromatic and aliphatic substituted propargylic alcohol acetates were demonstrated. The authors have performed mechanistic experiments, including control experiments and a nonlinear effect study, which reveals the vital role of the 2'-hydroxyl group of chiral aldehyde in this reaction. They also detect the key intermediates by HRMS analysis. The authors have proposed a possible reaction mechanism that shows a crucial role of ZnCl_2 in this catalytic process. The authors have also synthesized four stereoisomers of natural product NP25302. A similar methodology to synthesize chiral propargyl-functionalized amino acids has been demonstrated by Guo using Ni/Cu bimetallic system, though protected amino acids were employed. In fact, in the current method, the employed aldehydes are used as a transient protecting group, which requires an additional ZnCl_2 precursor. Moreover, the same group extensively demonstrated the original concept of "chiral aldehyde/metal combined catalysis" for the synthesis of diverse molecules (ref 36-38), and other groups have demonstrated the use of propargylic electrophiles in the reactions (ref 50-59). Considering all the precedents, the present work conceptually lacks the novelty to be considered for publication in the journal Nat. Commun.

Our response: Thanks for your comments very much. The value of this work was not only lie in the utilization of N-protected amino acid esters as reactants but the discovery of a novel asymmetric catalytic system consisting of a chiral aldehyde and a transition metal other than palladium. Because of the different chemical properties of palladium and nickel, this novel

catalytic system is anticipated to realize challenging organic transformations that could not be achieved by chiral aldehyde/palladium catalysis.

Numerous errors in the manuscript should be addressed:

Question 1. In the abstract and elsewhere, modify the sentences as ".....:the asymmetric propargylation reaction of amino acid esters with (not and) propargylic alcohol.....".

Our response and revisions: Thanks for your reminder very much. We corrected this issue as suggested.

Question 2. Reaction optimization: why does the product yield decrease upon increasing the concentration of TMG (Table S2, entry 5)? Similarly, in solvent screening, the reaction in toluene gave the 71% yield, while o-xylene and mesitylene gave only 4% yield (Table S7, entries 4-6).

Our response: Before the reaction started, the transition metal and ligand were mixed and stirred in solvent for the in situ preparation of an active nickel catalyst. We found the mixture became muddy and some unknown solid was precipitated once much more equivalents of TMG were used, or this reaction was carried out in xylene or mesitylene. By contrast, a clear solution was obtained when this reaction was carried out in toluene and with no more than 1.6 equivalents of TMG were used. We thought it was the difficulty in forming an effective transition metal catalyst that caused the great decrease in yields. Due to the lack of solid evidence, we did not give any description in our manuscript.

Question 3. What is "ent-3i"? Clearly define the same. Reaction time is not mentioned in Table 1 or the footnotes.

Our response and revisions: Thanks for your suggestion very much. 'Ent-3i' refers to the enantiomer of the chiral aldehyde catalyst **3i**. We gave the full description in our Revised Manuscript. The reaction time was added in the graphic of Table 1.

Question 4. Table 1, entry 22: reaction should have been checked with a Pd(0) precursor, as Ni(0) is used in the original reaction. Moreover, the formula of Pd(II) species in the footnotes of Table 1 is incorrect.

Our response and revisions: Thanks for your suggestion very much. We carried out this reaction under the promotion of chiral aldehyde/Pd(0) catalysis and no desired reaction occurred. We added this result in the footnote of Table 1. The wrong-spelled formula in the

original manuscript was corrected.

Question 5. Figure 4: Please mention the reaction partners from 9 to 10 and 10 to 11

Our response and revisions: Thanks for your suggestion very much, these issues were corrected as suggested.

Question 6. Page 2: “unique priority of chiral aldehyde”: the word “priority” seems odd here.

Our response and revisions: Thanks for your comment very much, we changed the word ‘priority’ to ‘properties’ in our Revised Manuscript.

Question 7. The authors should provide ¹⁹F NMR spectra for fluorinated compounds.

Our response and revisions: Thanks for your suggestion very much. ¹⁹F NMR spectra were provided for all of the fluoro-contained products in this revision.

Question 8. In Table 1, the molecular formula for chiral aldehyde 3g is incorrect.

Our response and revisions: Thanks for your reminder very much, we corrected this wrong-spelled molecular formula in this revision.

Question 9. On page no 6, line no 1, the spelling of propargylic is incorrect.

Our response and revisions: Thanks for your reminder very much, we corrected this wrong-spelled word in this revision.

Question 10. References 45 and 47 are the same.

Our response and revisions: Thanks for your reminder very much, we deleted the duplicated reference in this revision and re-adjusted the reference numbers.

3. Response and revisions to Reviewer #3

Main comments: Guo, Wen and co-workers described a chiral aldehyde/nickel catalytic system and its application in the asymmetric α -propargylation reactions of amino acid esters and propargyl alcohol derivatives. The development of novel combining catalytic systems from organocatalysts and transition metals is one of the most active research fields in asymmetric catalysis. Although chiral aldehyde catalysts exhibited powerful activation and excellent stereoselective-control abilities in the amino acids-involved reactions, examples of combining catalysis derived from chiral aldehydes and transition metals are very limited. Based on their research on the chiral aldehyde/palladium catalysis, the authors employed the transition metal nickel as catalyst partner for chiral aldehyde in this work. Due to the different chemical properties between palladium and nickel, chiral aldehyde/nickel catalysis is

anticipated to realize a wide range of new reactions that cannot be achieved by previously reported catalytic systems, such as the asymmetric propargylation reaction disclosed in this work. Furthermore, the methodology disclosed here has broad substrate scopes and excellent stereoselective outcomes, and the application of this methodology was well documented by the stereodivergent synthesis of natural product NP25302. So, this reviewer recommended this work to be accepted by Nature Communications. Before acceptance, the following issues should be addressed.

Question 1. Two chiral sources, the chiral aldehyde and chiral ligand, were used in this reaction. Ligand screening indicated that chiral diphosphine is necessary for this reaction, but no results to illustrate the necessity of chiral aldehyde. A combining catalytic system derived from achiral aldehyde and chiral palladium should be examined in this reaction.

Our response and revisions: Thanks for your suggestion very much. We tested two achiral aldehyde catalysts in this revision (Table 1, entries 21-22 in Revised Manuscript). Results indicated that both of them could not give satisfactory yields and enantioselectivities. So, the combination of a chiral aldehyde and a chiral nickel catalyst is necessary for this reaction.

Question 2. In the supplementary information, the HPLC spectrums of compounds 6 and 9 showed that there existed a lot of impurities. So these two compounds should be further purified and corresponding data of yields should be modified.

Our response and revisions: Thanks for your comments very much. We checked the ^1H NMR spectra of these compounds and found that the quality of these spectra could meet the publication requirement. So, the data of yields and enantioselectivities we reported were valid. Due to the low UV absorption of these compounds, the baseline noise and the UV absorption of some low-concentration impurities could not be well suppressed.

All of the above revisions were marked in red font in our Revised Manuscript.

Hopefully, this revised version of our manuscript can meet the high requirement of Nature Communications.

Sincerely yours,

Best wishes,

Dr. Guo

REVIEWERS' COMMENTS

Reviewer #1 (Remarks to the Author):

In their revised manuscript and response letter, the authors have addressed the main points raised by the reviewers to a good degree. Therefore, the reviewer would now support the publication of this paper in Nature Communications.

Reviewer #3 (Remarks to the Author):

This reviewer's concerns are addressed and I am now supportive of its publication in Nature Communications.